# Adenomyosis and Infertility—Review of Medical and Surgical Approaches

**DOI:** 10.3390/ijerph18031235

**Published:** 2021-01-30

**Authors:** Maria Szubert, Edward Koziróg, Olga Olszak, Klaudia Krygier-Kurz, Jakub Kazmierczak, Jacek Wilczynski

**Affiliations:** 1Clinic of Surgical and Oncologic Gynecology, 1st Department of Gynecology and Obstetrics, Medical University of Lodz, 90-419 Lodz, Poland; edward.kozirog@gmail.com (E.K.); klaudia.krygier89@gmail.com (K.K.-K.); j.kazmierczak87@gmail.com (J.K.); jrwil@post.pl (J.W.); 2Clinic of Perinatology, 1st Department of Gynecology and Obstetrics, Medical University of Lodz, 90-419 Lodz, Poland; olgaapietrzak@gmail.com

**Keywords:** adenomyosis, infertility, GnRH, in vitro fertilisation

## Abstract

The aim of this review is to clarify the relative association between adenomyosis and infertility and the possible treatment for an infertile patient. Although adenomyosis is detected more often in women of late reproductive age, its influence on pregnancy rates is important, especially considering the tendency to delay pregnancy among women in developed countries. In this article, we present a critical analysis of the literature data concerning the impact of adenomyosis on fertility. The possible effects of treatment on the pregnancy rate will also be discussed. We conducted a literature search; publications from Pubmed, Embase and Cochrane databases published from 1982 to 2019 were retrieved using terms ’adenomyosis and infertility’ and ’adenomyosis and pregnancy outcomes’, extensively studied in the aspects of diagnosis, pathogenesis of infertility and possible treatment methods. Molecular studies have given deep insight into the pathogenesis of adenomyosis in the recent few years, but there is a huge discrepancy between in vitro studies and praxis. Oral contraceptive pills, anti-prostaglandins, oral or parenteral progestins, danazol and gonadotrophin-releasing hormone (GnRH) analogues have all been used to control menstrual pain and menorrhagia in women with adenomyosis, but they temporarily suppress the menstrual cycle. Additionally, endometrial ablation and hysterectomy used to alleviate pain caused by adenomyosis exclude pregnancy planning. The development of imaging techniques—ultrasound and MRI—enables the diagnosis of adenomyosis with very high accuracy nowadays, but the methods of treatment mentioned above have not given satisfactory results in women planning pregnancy. For these patients, the high-intensity-focused ultrasound method (HIFU) and combined treatment before assisted reproductive techniques can prove beneficial in adenomyosis patients.

## 1. Introduction

The aim of this article was to provide readers with the newest information useful in the management of infertility in adenomyosis patients. Adenomiosis, well-described at the end of the 19th century, still remains a mysterious disease with severe implications on fertility. We conducted a literature search—publications from Pubmed, Embase and Cochrane databases published from 1982 to 2019 were retrieved using terms ‘adenomyosis and infertility’ and ‘adenomyosis and pregnancy outcomes’. Data of this search are provided in Figure 1. We focused on all aspects of infertility in adenomyosis: Symptoms, pathological background, diagnostics and possible management.

## 2. Definition and Symptoms

Adenomyosis is defined as an invasion of the endometrium into the uterine myometrium, which results in an enlargement of the uterus, formation of adenomyotic tumours, profuse menstrual and inter-menstrual bleeding and recurrent pain. Microscopically ectopic nonneoplastic, endometrial glands and stroma surrounded by the hypertrophic and hyperplastic myometrium are noted.

The prevalence of adenomyosis fluctuates between 5 and 70% [1]. Before the age of 40 years, the disease affects 2 in 10 women, whereas between 40 and 50 years, the incidence increases to 8 in 10 women [2]. However, the incidence of adenomyosis is difficult to establish due to the lack of a unified definition and diagnostic criteria based on noninvasive diagnostic tests [3]. There are no pathognomonic clinical features for adenomyosis, nor laparoscopic criteria that could be implemented for the diagnosis [4].

In fact, adenomyosis was previously diagnosed in premenopausal women only on the basis of pathological examination after hysterectomy [5,6]. Nowadays, the diagnosis is based on imaging techniques such as transvaginal ultrasound scan (US) and magnetic resonance imaging (MRI) [7]. In one third of cases, adenomyosis is asymptomatic. The most common clinical symptoms are menorrhagia (up to 50% of patients), dysmenorrhea (30%) and metrorrhagia (20%), with other medical conditions such as enlarged uterus and infertility [2,6].

Adenomyosis may be accompanied by other mild oestrogen-dependent benign disorders such as endometriosis (70%), uterine fibroids (50%) and endometrial hyperplasia (35%). In the retrospective analysis of 945 patients who underwent hysterectomy, a significant positive correlation was found between the progression of adenomyosis and history of prior abortion, history of previous pregnancies and occurrence of leiomyoma. By contrast, there was no correlation with smoking, normal delivery, caesarean section, endometrial hyperplasia or ovarian endometriosis [8].

## 3. Pathogenesis

The pathogenesis of adenomyosis is still unclear. It may develop de novo from a metaplastic transformation of the embryological pluripotent mullerian remnants. The second theory, suggested by Bergeron et al., is an invasion of the basal endometrium into the myometrium through an altered or absent JZ (junctional zone—the area representing the internal myometrium) [9]. The invagination and intramyometrial spreading may be due to higher oestradiol receptor expression in the adenomyotic foci. Another hypothesis formulated by Leyendecker et al. concerns ’auto-traumatisation’ of the uterus which leads to the TIAR (Tissue Injury And Repair) mechanism as the main cause of adenomyosis [10]. According to the authors, high intrauterine pressure, especially during menstruations, can cause a rupture of the archimyometrium mainly in the cornual region of the uterus. The TIAR mechanism causes a vicious cycle of hyper-estrogenic activity and expression of P450 aromatase. The TIAR mechanism can also occur after multiple D and C procedures (dilatation and curettage).

Many macromolecules such as hormones, cytokines and antigens might play a role in the pathogenesis of adenomyosis. The hyper-estrogenic environment promotes IL-10 expression. IL-10 may have an influence on the maintenance of host immunosuppression, augmenting the growth of adenomyotic foci. IL-1 and IL-6 mediate the inflammatory response by COX2- and PGE2-dependent pathways. An abnormal congestion of reactive oxygen species is also observed in adenomyosis, with the presence of ROS (reactive oxygen species), the nitric oxide derivative, being particularly abundant [11].

Moreover, the myometrium itself may be involved in disease development by influencing local biochemical factors like cytokines and oestrogens, which play a role in smooth muscle metaplasia and/or the trans-differentiation of fibroblasts into myofibroblasts [12].

A new direction of research is the expression of messenger RNA (mRNA) and long noncoding RNAs (lncRNAs) in adenomyosis. Lnc RNAs, earlier considered as transcriptional noise due to their low level of expression, are now being studied for a variety of biological functions, such as immune response and cell differentiation. Zhou et al. investigated the expression patterns of lncRNAs in human adenomyosis tissue. In microarray analysis, they showed that uc004dwe.2 lncRNA, which may affect the angio- and lymphangiogenetic function of NRP2 (neuropilin 2), is overexpressed in the ectopic endometrium compared to eutopic endometrium [13]. The limited understanding of lncRNA function in adenomyosis requires further research.

## 4. Diagnostics

Adenomyosis occurs in two different forms: Diffuse and focal, usually observed during transvaginal US examination. Focal forms of adenomyosis are presented as pseudo-widening, adenomyoma and haemorrhagic cysts [1]. So far, various systems have been proposed to describe adenomyosis [14].

### 4.1. Ultrasound

Ultrasound is the first-line imaging tool in the infertile patient. General diagnostic US features of the two forms of adenomyosis are presented in Table 1.

A characteristic symptom observed during ultrasound examination is the general enlargement of the uterus, which usually reaches 12 cm in length and cannot be explained by the presence of uterine fibroids, and is a characteristic finding in US examination. In addition, a sign called ‘the question mark form of the uterus’ (when the uterine corpus is flexed backwards, and the cervix is directed anteriorly towards the urinary bladder) is related to high sensitivity and specificity (92% and 75%, respectively) of US [1]. The transformation or the junctional zone is a layer that appears as a hypoechoic halo surrounding the endometrial layer [11]. Thickening of the junctional zone is a visible US sign of endometrial invasion into the myometrium. In a patient with suspected adenomyosis, a JZ thickness greater than 12 mm strongly suggests the presence of an affected myometrium, while a JZ thickness between 8 and to 12 mm may suggest solely adenomyosis, but other signs are required to confirm its presence [2].

Indeed, 2D—and 3D—TVS allows easy recognition of diagnostic signs (Figure 2). Eisnenberg et al. observed a very high overall prevalence (89.4%) of sonographic signs of adenomyosis in women undergoing laparoscopic surgery for endometriosis, much higher than in the control group [6]. In this study, a higher risk of infertility was correlated with a greater number of sonographic signs of adenomyosis. Additional examinations such as MRI are useful in identifying the variation in JZ and to exclude concomitant disorders [1].

### 4.2. Hysteroscopy

In a specific group of patients (mainly those with an abnormal uterine bleeding pattern), hysteroscopy can be a valuable diagnostic technique which, on the one hand, enables direct visualisation of the uterine cavity, and, on the other hand, enables the collecting of material for histopathological examination. Although visual inspection does not allow a definite diagnosis, a number of features have been identified that may indicate the presence of adenomyosis: Pronounced hypervascularisation on the endometrial surface, an irregular endometrium with small openings, the so-called strawberry pattern of the endometrium and fibrous and haemorrhagic cystic lesions. More information can be obtained from the behaviour of the uterine muscle during the biopsy with a diathermy loop resectoscope. The presence of adenomyosis may be indicated by an irregular sub-medial myometrium, a distortion of the normal myometrial architecture noticeable during resection and the presence of intramural endometriomas [15].

### 4.3. MRI (Magnetic Resonance Imaging)

Pelvic MRI is the reference standard for the noninvasive detection of the adenomyosis in patients with infertility [16]. However, it requires expensive equipment and extensive knowledge and experience while assessing images. One should keep in mind that MRI better predicts adenomyosis while performed in the secretory phase of the menstrual cycle. The agreement is that a junctional zone of more than 12 mm in width is strongly associated with the disease. When the JZ measures 8–12 mm, the following key features should be assessed: More than 5 mm difference between the maximum and minimum thickness of the JZ, poor definition of borders of the JZ or high signal intensity foci in the myometrium [17].

### 4.4. Histological Evaluation

The key features in the diagnosis are endometrial glands within uterine muscles. The glands are circular in shape, filled with blood and surrounded by endometrial stroma. The epithelium of the glands could be pseudostratified, may have mitoses, and are densely packed spindle cells without nuclear atypia. Between epithelial cells, macrophages loaded with hemosiderin are observed. The surrounding myometrium is usually hyperplastic. There should be at least 2.5 mm distance between the endomyometrial junction and the adenomyotic gland to establish proper diagnosis of adenomyosis [9]. A histopathological report is obviously not required to treat qualified patients for infertility.

## 5. Biological Influence of Adenomyosis on Fertility—Possible Mechanisms

Recent studies show that adenomyosis negatively affects in vitro fertilisation, pregnancy and the live birth rate, as well as increases the risk of miscarriage. In addition, adenomyosis enhances the risk of obstetric complications, such as premature birth and preterm rupture of the amniotic membranes [18,19].

Fertility in adenomyotic patients could be disturbed by various mechanisms. Abnormal utero-tubal gamete and the embryo transport and disruption of endometrial function and receptivity have been described [2]. An enlarged uterus, anatomical distortion and intramural adenomyoma can all influence the shape of the uterine cavity. It may have a negative impact on sperm migration, embryo transfer and implantation potential [2,20]. The researchers suggested an association between spontaneous abortion and JZ function [21]. Chiang at al. observed a comparable dependence [22]. An average JZ greater than 7 mm was correlated with higher implantation failure [23].

Hyperactivity of the myometrium is also observed in adenomyosis. Changes in myocytes are also found on the cellular level—calcium circulation is distorted, which implies irregular muscle contractions, dysfunctional uterine hyperperistalsis with increased intrauterine pressure and the development of hyperplastic myometrial tissue. The thickening of the junctional zone is a visible sign of endometrial invasion into the myometrium. Alerted myometrial contractility may impair sperm progression towards the peritoneal opening of the tubes [2].

Distortion of the uterine cavity can be visualised in hysterosalpingography (HSG), and occurs in 78% of patients with diffuse adenomyosis and 54% cases of focal adenomyosis compared to 37% of women without adenomyosis. These findings may suggest the association between adenomyosis and the probability of abnormal utero-tubal transport [5].

Adenomyosis-associated changes may also worsen endometrial receptivity [24]. Endometrial receptivity is defined as physiological molecular and histological phenomena occurring during a restricted time of the menstrual cycle, making the uterus exclusively receptive to blastocyst attachment and implantation (so-called implantation window).

Evidence of reduced endometrial receptivity and impaired decidualisation in adenomyosis was found at the molecular level. Abnormal function of the implantation-associated molecules such as HOXA10, LIF, MMP, IL-6, IL -10, cytochrome P450 and RCAS1 has been described [25].

The decreased expression level of HOXA10 genes in the secretory phase endometrium appears to be involved in impaired implantation in women with adenomyosis. Similarly, a deregulation of leukaemia inhibitory factors (LIF) in uterine flushing fluid during the implantation window has been reported in adenomyosis [5,26,27]. Jiang et al. reported the down-regulation of the NR4A receptor and FOXO1A in adenomyotic tissue, which leads to incorrect decidualisation [28]. Whether these changes can be restored by the progestins given during the implantation window remains unknown due to the lack of conclusive data in humans [29,30].

Certain cell adhesion molecules, such as integrins, are also extensively studied in adenomyosis. Integrins are transmembrane receptors, which activate signalling pathways and mediate cellular signals such as regulation of the cell cycle. Integrins are responsible for endometrial receptivity. Abnormal expression of both integrin β-3 and OPN mRNA (osteopontin, responsible for the trophoblast-endometrium interaction) is found in adenomyosis patients, and it is suggested that this abnormal expression may be responsible for in vitro fertilisation (IVF) failure despite good embryo quality [31]. Integirn β3 together with osteopontin (OPN) are involved in cell–cell interactions, and their proper functioning is inevitably related to uterine receptivity. In the endometrium of adenomyotic patients, the levels of β3 and OPN were statistically lower compared to nonadenomyotic controls [25,27]. The influence of medical treatment (gonadotrophin-releasing hormone (GnRH) analogues, ovarian stimulation) on integrin expression and endometrial receptivity has so far been studied in animal models and could only partially be conclusive for human pathology [32,33].

It is a well-known fact that chronic inflammation has a negative impact on fertility [34,35,36,37,38]. In the case of a patient with adenomyosis, an increased expression of IL-1b and CRH (corticotrophin-releasing hormone) in the eutopic endometrium was observed [34]. In addition, the presented data showed differences in both cellular and humoral immunity in the eutopic endometrium of an adenomyotic uterus compared to the unaffected control [35]. Ishikawa et al. reported an increased inflammatory response in the endometrium due to the presence of a higher expression of pro- and anti-oxidative cytokines like Cu, Zn-SOD and Mn-SOD [36]. Other authors confirmed these findings by investigating the nitric oxide (NO) concentration in endometrium, macrophage activation, Il-6 and neurotrophins [25,37]. NO is involved in modulating uterine contractility during pregnancy and relaxing vascular smooth muscles. An abnormal high level of free radicals such as nitric oxide has a negative impact on sperm transport, implantation and decidualisation [38].

## 6. Impact of Adenomyosis Treatment on Fertility and Implication for Clinical Practice

As in the case of endometriosis, the management strategy of adenomyosis depends primarily on the presented symptoms. The pharmacological treatment of adenomyosis is similar to that of endometriosis, but the data concerning its influence on fertility remain inconclusive. Final treatment with a hysterectomy is the most effective way of achieving symptoms control and provides high satisfaction rates. However, for obvious reasons, it is unacceptable for women wishing to have children. Fertility-saving treatment has variable success rates for both pain and bleeding. Moreover, some of the available nonsurgical management methods severely interfere with fertility. The methods accepted for treatment of adenomyosis are listed in Table 2 and described in detail below:

### 6.1. Surgical Methods

Endo-myometrial resection is effective and indicated in patients with the disease limited to the endo-myometrial junction and allows the reduction of heavy menstrual bleeding [39]. However, in patients who desire pregnancy, endo-myometrial resection is contraindicated [40]. Destruction of the endometrium together with JZ can cause serious complications in patients who managed to conceive, such as miscarriage, preterm labour and placentation complications. An unexpectedly high rate of pregnancy complications is reported in a systemic review by Kohn et al. [41]. Embolisation has also been described as an effective treatment of symptoms resulting from adenomyosis. This endovascular procedure causes the closing of vessels that supply the uterus. Premature ovarian insufficiency (POI) is mentioned as a consequence of embolisation and its rate should not be underestimated. It can affect both hormone production and ovarian reserve, leading to premature and iatrogenic amenorrhoea and infertility. Endometrial receptivity is also diminished after this procedure. Therefore, it should be contraindicated in women planning pregnancy but is useful in the post-reproductive age [42]. The high-intensity focused ultrasound method (HIFU) uses the thermal effect of the ultrasound beam, which causes coagulative necrosis within the targeted adenomyotic lesion. The lesion should be clearly visible in ultrasound or under MRI so that the beam could be precisely directed. This means that this method will be unsuitable for the diffuse form of adenomyosis. After the procedure, patients can attempt to conceive much earlier than after surgical treatment, but the exact time of delay in conception is unknown. Zhang et al. published data which indicate that the rate of uterine ruptures during pregnancy or delivery is lower than after classical surgical methods [43]. In a retrospective analysis of HIFU-treated patients, Zhou et al. found that 54 patients out of 68 conceived after HIFU, delivering 21 babies [44]. Although the miscarriages rate appears to be quite high, other severe complications like uterine rupture did not occur. It is suggested that in the HIFU method, myometrium around the adenomyotic lesion is intact, so there is no scar on the uterine wall. In classical surgical methods, the removal of significant amounts of myometrium with the adenomyotic lesion may result in a reduction in myometrial capacity of the uterus and in the formation of uterine scars. The first one can cause a lack of susceptibility of the uterus to grow during pregnancy, while the latter can cause a risk of uterine rupture. Otsubo at al. presented results of fertility-saving surgical excisions of diffuse adenomyotic foci and suggested that preservation of a 9 to 15 mm thickness of the uterine wall after excision with classical surgical methods is safe for future pregnancies [45].

Electrocoagulation has also been applied to focal or diffuse disease. However, the main disadvantage of electrocoagulation is that it may be less accurate than surgical excision, as well as poorly controlled during the procedure. Younes et al. concluded in the recent meta-analysis that conservative surgery in adenomyosis could improve fertility in some patients, but the rate of resulting successful pregnancies varied between surgical centres [46].

### 6.2. Pharmacological Methods

Nonsteroidal anti-inflammatory drugs (NSAIDs) are widely used in endometriosis-associated pain, but there are only a few randomised trials in endometriosis and none of them were performed in adenomyosis [47]. Their impact on fertility is negative; they can cause a delay in ovarian follicle rupture, but there is some evidence that NSAIDs can be used as a co-treatment in the IVF procedure [48]. OCs (oral contraceptives) are used in the treatment of adenomyosis to reduce the menstrual bleeding by decidualisation and subsequent endometrial atrophy. Patients with dysmenorrhoea and menorrhagia may benefit from the resulting amenorrhoea, which may alleviate symptoms. Medical therapy with OCs enables satisfactory, long-term pain control in two-thirds of women with symptomatic endometriosis or adenomyosis. However, there is no published information on the impact of OCs therapy on the subsequent fertility improvement [49]. GnRH analogues have been used to induce a constant hypoestrogenic state in women with histologically proven adenomyosis [40]. Although their use is accepted in patients with dysmenorrhoea due to adenomyosis, there are still very few data on their impact on future fertility. In one small case series published by Huang et al., authors did not show an improvement in fertility after the GnRH analogue management combined with conservative microsurgery [50]. Other results found in the literature are also vague—as presented in Table 3. An antiproliferative and anti-inflammatory effect of progestins was the basis of their use in adenomyosis treatment [51]. It has been found that they are at least partially effective in controlling pain symptoms associated with adenomyosis. They were able to reduce uterine volume and affect abnormal uterine bleeding, but their influence on fertility is poorly documented in either short case series or case reports [52,53,54].

## 7. IVF Outcome in Adenomyosis

The results of studies devoted to the efficacy of assisted reproductive techniques, such as in vitro fertilisation and embryo transfer (IVF-ET) and intracytoplasmic sperm injection (ICSI), on pregnancy rates in patients with adenomyosis showed conflicting results. Mijatovic et al. did not observe significant differences in clinical pregnancy rates in patients with adenomyosis in infertile women with proven endometriosis who were pretreated with long-term GnRH-agonist compared to controls [55].

Additionally, Thalluri and Tremellen noticed statistically significantly lower clinical pregnancy rates in patients subjected to IVF-ET with adenomyosis despite the GnRH stimulation protocol [56]. Contrarily, Costello et al. did not observe impaired fertility among patients with adenomyosis who received GnRH during IVF-ICSI [57].

Finally, Vercellini et al. published a meta-analysis confirming the negative effect of adenomyosis on the IVF-ICSI outcomes, impairing the rate of clinical pregnancy and implantation and increasing the risk of early pregnancy loss [58].

Adenomyosis was a risk factor reducing the rate of implantation and clinical pregnancy, as well as increasing the risk of early pregnancy loss. The presented data are generally inconclusive—see Table 3. The cohorts of patients were heterogeneous; patients were of different ages and differed in the clinical presentation of adenomyosis and concomitant endometriosis. The classification of patients to an appropriate group for studying purposes is difficult, which is connected to the lack of standards in the diagnosis of adenomyosis.

## 8. Conclusions

There is no specific treatment for patients with adenomyosis who want to retain their uterus or wish to preserve fertility [72]. Sometimes, combined treatment can be proposed: Laparoscopy, GnRH treatment and in vitro fertilisation [73].

When comparing pharmacological and surgical treatment, the latter appears to be more effective but some details are unclear, i.e., how long pregnancy should be delayed after treatment and whether hormone treatment after surgery improves fertility outcome. Despite many studies on the pathogenesis of fertility failure in adenomyosis, their results are not correlated with treatment. Thus, it is of great importance to explore new, more effective, safe and less invasive managing strategies in women with infertility due to adenomyosis.

## Figures and Tables

**Figure 1 ijerph-18-01235-f001:**
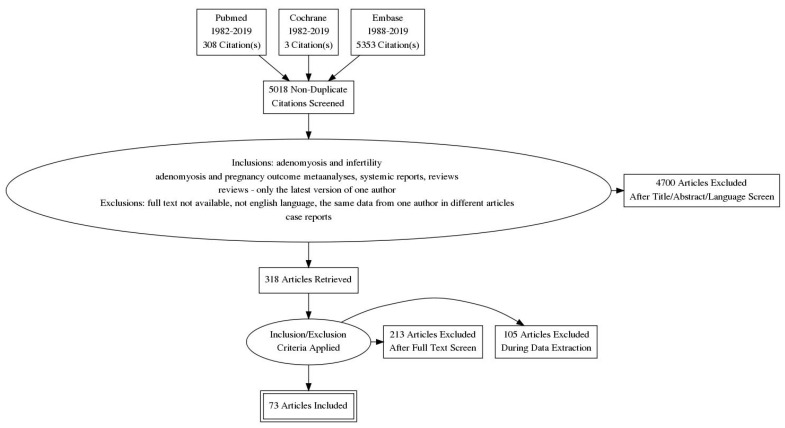
Process of data extraction.

**Figure 2 ijerph-18-01235-f002:**
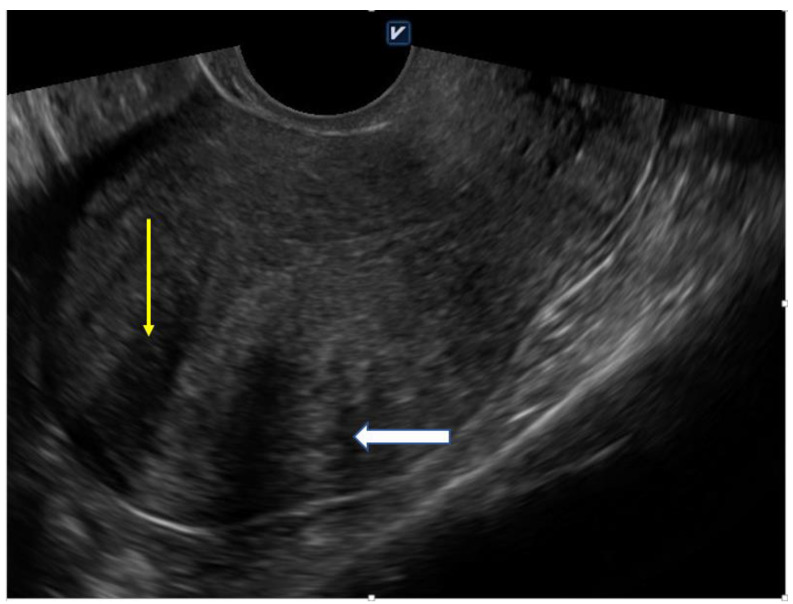
Globally enlarged uterus with heterogenic myometrium (yellow arrow) and asymmetric thickeness (white arrow); 41 year-old nulliparous woman with 5 years of infertility anamnesis (own material).

**Table 1 ijerph-18-01235-t001:** Sonographic features of diffuse and focal adenomyosis.

Diffuse Adenomyosis	Focal Adenomyosis
globally enlarged uterus	focal disturbances in myometrium layer
asymmetric thickness anterior and posterior wall = pseudo-widening sign	sometimes focal form diagnosed as intramural myoma
cystic myometrium (cystic anechoic spaces)	anechoic cysts
junctional zone not clearly visible, thickening of the JZ	
heterogeneous echogenicity of the myometrium	

**Table 2 ijerph-18-01235-t002:** Adenomyosis treatment.

Pharmacological	Surgical
Anti-inflammatory drugs	Endo-myometrial ablation
Oral contraceptives	High-intensity focused ultrasound
GnRH	Ablation
progestins	Electrocoagulation of adenomyosis foci
	Resection of adenomyosis foci
	Hysterectomy

**Table 3 ijerph-18-01235-t003:** Fertility outcomes and impact of different procedures on adenomyosis symptoms.

Author	Treatment	Patients	Results		
		N	Fertility	Bleeding	Pain
Kwack et al. 2018 [59]	conservative adenomyomectomy with TOUA (transient occlusion of uterine arteries)	116	5/116 conception by natura l5/11 conception by ART 7 live births	menorrhagia: 52/116 complete remission 53/116 partial remission	dysmenorrhea: 98/116 complete remission 18/116 partial remission
Al Jama et al. 2016 [60]	treatment with Gn-RH agonist	22	3/22 pregnancies 1/22 live birth	improvement in dysmenorrhea and menorrhagia was noted at the 6- and 12-month follow-up visits in both groups	
	combined conservative surgery and Gn-RHa therapy	18	8/18 pregnancies 6/18 live births		
Saremi et al. 2014 [61]	resection of adenomatosis lesions with a thin margin after sagittal incision in the uterine body	103	14/70 conception by ART 7/70 conception by natural 16/70 live births	decrease of 65% in the number of patients with a heavy bleeding pattern;	decrease of 41% in the number of patients with dysmenorrhoea symptoms;
Kishi et al. 2014 [62]	laparoscopic adenomyomectomy with laser	102	conception by natural: 16/75 (<40 y) and 0/27 (40 or more y) conception by ART: 15/75 (<40 y) and 1/27 (40 or more y) delivery: 26/75 (<40 y)	no data	no data
Chang et al. 2013 [63]	ultramini- or mini-laparotomy conservative surgery and Gn-RHa therapy	56	23/56 pregnancies 15/56 live births	no precise data	VNRS-6 (six-point verbal numeric rating scale) baseline of 3.96 ± 0.41 to 0.32 ± 0.46 1st year, 0.68 ± 0.78 2nd year 1.27 ± 1.22 3rd year,
Dai et al. 2012 [64]	local excision of adenomyoma at laparotomy	86	2/86 pregnancies	no data	alleviation of dysmenorrhea -12 months after treatment: >80% reduction in 77/79 (97.5%); 50–80% reduction in 2/79 (2.5%); -24 months after treatment: >80% reduction in 45/48 (93.8%); 50–80% reduction in 3/48 (6.2%);
Huang et al. 2012 [50]	excision of the adenomyosis tissue using a microsurgical technique and a six-month course of GnRHa therapy	9	6/18 conception by ART 3/18 conception by natural 2/18 live births	no data	pain score 4.7 ± 0.5 before treatment 0.33 ± 0.5 after 3 months 1.0 ± 0.9 after 12 months
Osada et al. 2011 [65]	adenomyomectomy with a triple-flap method, without overlapping suture lines	104	4/26 conception by natura l12/26 conception by ART 14 live births	VAS (visual analogue scale) hypermenorrhoea, 10 pre-surgically 3.27 ± 2.17 at 3 months, 2.89 ± 1.77 at 6 months, 2.63 ± 1.3 at 1 year, 2.87 ± 1.77 at 2 years post-surgery	The VAS findings (dysmenorrhoea, 10 pre-surgically) 1.61 ± 1.43 at 3 months, 1.54 ± 1.62 at 6 months, 1.44 ± 1.65 at 1 year, 1.67 ± 1.79 at 2 years post-surgery.
Takeuchi et al. 2010 [66]	laparoscopic enucleation of juvenile cystic adenomyoma	9	2/3 pregnancies	no data	dysmenorrhea 8–10 on the VAS before the surgery, decreased to 2 by 6 months after
Nishida et al. 2010 [67]	adenomyomectomy with unilateral salpingectomy	44	1/16 live births	reducing menstrual blood loss, no quantitative data	dramatic relief from dysmenorrhea, no quantitative data
Hadisaputra et al. 2006 [68]	laparoscopic resection +GnRH analogue after surgery	10	3/10 pregnancies	no change in the symptom of menorrhagia	75% reduction in dysmenorrhea after treatment
	myolysis +GnRH analogue after surgery	10	2/10 pregnancies	no change in the symptom of menorrhagia	58.31% reduction in dysmenorrhea after treatment
Rajuddin et al. 2006 [69]	laparotomic resection	32	3/32 pregnancies 2/32 live births	after intervention, 28/32 experienced disappearance of symptoms (dysmenorrhea, pelvic pain, menorrhagia, dyspareunia), while 4/32 had remaining symptoms;	
	treatment with aromatase inhibitor of anastrozole	23	2/23 pregnancies 1/23 live births	after therapy 14/23 experienced a disappearance of symptoms, while 9/23 had remaining symptoms;	
Takeuchi et al. 2006 [70]	laparoscopic adenomyomectomy and hysteroplasty	14	2/14 pregnancies	all 8 cases of polyhypermenorrhea improved, no precise data available	dysmenorrhea—VAS during menstruation decreased from 10 before operation to 2.5 after operation.
Fujishita et al. 2004 [71]	classical reduction surgery	5	0/5 pregnancies	2/5 relief of menorrhagia and dysmenorrhea	
	transverse H incision method and the reduction surgery	6	1/6 pregnancies	3/6 relief of symptoms	

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
