# Peer review of "Adenomyosis and Infertility—Review of Medical and Surgical Approaches"

_ijerph, 2021, doi:10.3390/ijerph18031235_

Round 1

Reviewer 1 Report

I read with great interest the Manuscript titled “Adenomyosis and infertility – review of pharmacological and surgical approaches” (ijerph-1063613), which falls within the aim of International Journal of Environmental Research and Public Health.     

In my honest opinion, the topic is interesting enough to attract the readers’ attention. Nevertheless, the authors should clarify some methodological points and improve the discussion citing relevant and novel key articles about the topic.

Authors should consider the following recommendations:

  • The Manuscript should be further revised by a native English speaker.
  • I would like to suggest to the authors to stress the role of RNA in the pathogenesis and progression of adenomyosis, similarly to what happens in endometriosis and other endometrial pathologies. Authors should discuss this point, referring to: PMID: 26662114; PMID: 30037059
  • I recommend underlining the utility of hysteroscopy for the diagnosis of adenomyosis in the cases of female infertility. Authors should stress this point, referring to: PMID: 28852646; PMID: 33261529

Author Response

Dear Reviewer,

Thank you very much for your valuable comments. We implemented a lot of crucial changes in to the manuscript, both concering language as well as providing new data on imaging techniques and treatment.

Authors should consider the following recommendations:

  • The Manuscript should be further revised by a native English speaker. - done.
  • I would like to suggest to the authors to stress the role of RNA in the pathogenesis and progression of adenomyosis, similarly to what happens in endometriosis and other endometrial pathologies. Authors should discuss this point, referring to: PMID: 26662114; PMID: 30037059. We revised articles about role of RNA in adenomiosis and implemented comments in the lines: 108-116.
  • I recommend underlining the utility of hysteroscopy for the diagnosis of adenomyosis in the cases of female infertility. Authors should stress this point, referring to: PMID: 28852646; PMID: 33261529. Hysteroscopy has been described in lines: 156-169.

Reviewer 2 Report

First of all, I think the title"pharmacological" is inappropriate. Rather, I think it is appropriate to change "pharmacological"

-> " medical"

The conclusion in abstract is very poor.
I don't know what the writer is suggesting.

line 41 the range of the prevalence of adneomyosis is too wide, so plaease check it again.

line 83 oestrogens -> estrogens

picture It must be to chagne up and down.

with this publication, the authors want to provide additional reference on the subject and pleasse describe in more detail what needs to be said with more emphasis.

Author Response

Dear Reviewer,

Thank you for your revision. Please see my answer to your valuable comments below – marked yellow:

First of all, I think the title"pharmacological" is inappropriate. Rather, I think it is appropriate to change "pharmacological"

-> " medical" – Title has been changed

The conclusion in abstract is very poor.
I don't know what the writer is suggesting.

Abstract has been modified.

line 41 the range of the prevalence of adneomyosis is too wide, so plaease check it again.

It has been checked and the prevalence has been cited correctly.

line 83 oestrogens -> estrogens The change has been implemented.

picture It must be to chagne up and down. The change has been implemented

with this publication, the authors want to provide additional reference on the subject and pleasse describe in more detail what needs to be said with more emphasis.

We implemented a lot of crucial changes in to the manuscript, both concering language as well as providing new data on imaging techniques and treatment.

Reviewer 3 Report

This review article aims to clarify the relative association between adenomyosis and infertility and possible treatment methods in infertile patients. It utilized a literature search on Pubmed, Embase and Cochrane databases from 1982 to 2019. However, the current manuscript failed to highlight any new innovation or finding to help expand the current knowledge. Else, it would be similar to those review articles currently available. The author should digest further on the findings for more points in the discussion. Moreover, the comments are provided for the authors to possibly considered for improving the value of this article.

Major comments:

  1. Although this was a narrative review, it would be expected of a Prisma flowchart to illustrate the search results to be included in this review from the databases being searched.
  2. Toward the discussion on diagnostics, please include the section for histological evaluation. Similarly, why was the routine clinical diagnostics by MRI not discussed?
  3. In surgical methods, why was hysterectomy not being discussed? However, it was included in Table 2.
  4. Toward pharmacological treatment, it would be expected to have a discussion on the use of anti-inflammatory drugs (e.g. ibuprofen, Advil, Motrin IB, others) to control the pain, however this was missing.
  5. In the discussion on the effect of adenomyosis on IVF outcome, the author has not acknowledged the suggestion of combination treatments.
  6. Page 7 Line 256: This was not depicted correctly. In fact, the finding of Mijatovic V et al. actually showed pretreatment with long-term GnRHa did not affect the IVF outcome for patients with adenomyosis.
  7. How was the information selected for illustration for Table 3? Obviously, there is some critical findings not presented in the Table, but available partially in the text.
  8. Lastly, there is a major concern with the English in this manuscript, which would need to be edited by a professional writer.

Minor comments:

  1. Page 3 Line 106: Please start the sentence with Eisnenberg et al. and add the reference at the end of the sentence.
  2. Page 3 Line 112: What does Fot. 1. (near hear) mean? I reckoned this might be the figure title in Polish. Please clarify.
  3. Page 5 Line 170: please add the reference to the sentence “It is a well-known fact that chronic inflammation has negative effect on fertility.
  4. Figure 1: please add a figure title
  5. Figure 1: please remove (own material – Maria Szubert)
  6. Table 1: please state the fullname for the abbreviated term “JZ”

Round 2

Reviewer 3 Report

The authors have responded to all the questions raised. The manuscript has now been improved greatly.